# Predictors of neonatal hypothermia within six hours of birth and exploring preventive practices among post-natal mothers in Kilimanjaro region: Explanatory sequential mixed method protocol

**Emmanuel Daniel**[1]☯*, **Saada A. Seif**[2]☯, **Walter C. Millanzi**[3]☯

**1** Department of Clinical Nursing, School of Nursing and Public Health, The University of Dodoma, Dodoma City, Tanzania, **2** Department of Public Health and Community Nursing, School of Nursing and Public Health, The University of Dodoma, Dodoma City, Tanzania, **3** Department of Nursing Management and Education, School of Nursing and Public Health, The University of Dodoma, Dodoma City, Tanzania

☯ These authors contributed equally to this work.
* kijasawa@gmail.com

## Abstract

### Background

Neonatal hypothermia is a worldwide problem that can lead to a high impact on neonatal health outcomes if appropriate thermal care measures are not implemented. Tackling neonatal hypothermia from the time of delivery with appropriate thermal care measures will increase neonatal survival and decrease complications related to hypothermia.

### Objective

The study aims to determine the predictors of neonatal hypothermia within six hours of birth and explore its preventive practices among postnatal mothers.

### Methods

This is a mixed-method sequential explanatory study. The first phase will be a quantitative study with a hospital-based analytical cross-sectional design. 325 neonates and their mothers will be randomly selected through a 4-stage sampling technique. The data will be collected using a structured questionnaire, checklist and documentary review. Descriptive and inferential statistics will be used to analyse the data. The second phase will be a descriptive qualitative study involving postnatal mothers who participated in a quantitative study. The data will be collected via in-depth interviews, and thematic analysis will be used to analyse the data. The findings of quantitative and qualitative studies will be triangulated in the discussion.

**Data Availability Statement:** No datasets were generated or analysed during the current study. All

relevant data from this study will be made available upon study

**Funding:** The author(s) received no specific funding for this work.

**Competing interests:** The authors have declared that no competing interests exist.

## Discussion

This study will provide a wide understanding of neonatal hypothermia in the region which will help healthcare providers who are caring for neonates to be informed about the current situation, evaluate the care they provide, and find the best way to adhere to thermal care measures. Moreover, the practices of postnatal mothers will be known, which will help to develop tailored interactions to address this problem.

## Introduction

Neonatal hypothermia is a worldwide concern because it contributes to neonatal mortality, especially when it is concurrent with other neonatal problems. Neonatal hypothermia refers to a body temperature less than 36.50˚C and is further classified into three main classifications: mild hypothermia (360˚C-36.40˚C), moderate hypothermia (32.0˚C-35.90˚C), and severe hypothermia <32.0˚C [1]. Globally, neonatal hypothermia remains a major challenge to both developed, lower- and middle-income countries facing high neonatal mortality rates, as most of the previous studies that have been conducted in developed counties and Africa have reported that an increase in neonatal hypothermia is associated with neonatal mortality [2–6]. Preventing neonatal hypothermia is the key aspect of increasing neonatal survival and achieving Sustainable Development Goal Number 3.2 to prevent neonatal deaths by reducing neonatal mortality from 2030 to 12 per 1,000 live births [7].

The survival of neonates critically depends on maintaining normothermia after birth, which requires appropriate thermal care interventions, as during delivery, a neonate's body tries to adjust to the extrauterine life, as during this period, neonates can lose heat through evaporation, conduction, convection and radiation, which predisposes them to hypothermia [8, 9]. This also signifies the importance of thermal care after delivery.

The prevalence of neonates with hypothermia in tropical countries ranges from 8 to 92% across different studies that have reported neonatal hypothermia, and in hospital settings, it ranges from 32 to 85% among low- and middle-income countries. This variation in prevalence is due to measurement methods, climate, study design, population, type of collection tool, and definition of neonatal hypothermia [10]. In East Africa, neonatal hypothermia is still a challenging condition despite being managed with low resources and low cost. According to a study performed in East African countries, the pooled prevalence of neonates with hypothermia was 57.2% [11], which indicates that a problem still exists. Moreover, in Tanzania, two studies that were conducted in Dar Es Salaam showed that the percentages of neonates with low body temperature were 25.6% and 33%, respectively [12, 13]. Moreover, postnatal mothers play an important role in neonates' lives. Their practices will have an impact on the existence of the problem, as previous studies have reported that postnatal mothers did not practice skin-to-skin contact, neonates were soaked in water in the first 24 hours after delivery, and late initiation of breastfeeding was reported, all of which influence neonatal hypothermia [14, 15].

Protecting neonates against hypothermia, the World Health Organization introduced the thermal care practical guideline, which comprises ten steps: a warm delivery room that should be maintained at 25˚C for mature neonates and 26–32˚C for premature neonates; immediate drying and wrapping; late initiation of breastfeeding; lack of skin-to-skin contact; bathing of neonates after 24 hours; postponing weighing neonates for 60–90 minutes post-delivery; good clothing and bedding; putting together postnatal mothers with their neonates; warm transportation; warm resuscitation; and training and raising understanding [1], but there is still

suboptimal care in the region and little known what mothers do to protect their neonates to prevent neonatal hypothermia [16, 17]. This indicates that more efforts are required to explore this problem.

Neonatal hypothermia is influenced by delayed initiation of breastfeeding within one hour after birth, lack of skin-to-skin contact, prematurity, low APGAR score, resuscitation history, low birth weight, nighttime delivery and bathing within 24 hours [18, 19]. Nevertheless, it is not apparent whether the same factors are responsible for hypothermia in our study. However, thermal care is still a significant obstacle to neonate survival in lower- and middle-income countries such as Tanzania since neonatal mortality is still 24 per 1,000 live births [20], which indicates a double increase instead of a decline that does not align with SDG 3.2. Although addressing neonatal hypothermia may help achieve this objective, it has been a neglected problem. Nevertheless, women from different cultures have different ways of caring for children to prevent hypothermia, and some practices may be detrimental because they are not scientifically based, such as oil massage of the skin after birth [18]. Moreover, a study conducted in India showed bathing neonates soon after delivery and not drying the neonates immediately after delivery were the risk factors for neonatal hypothermia while to protect neonates against hypothermia, mothers used warm blankets, or clothes in multiple layers [21]. Similarly, the study conducted in 11 European countries showed that plastic bags, caps exothermic heat and trans warmer mattresses were effective methods for preventing hypothermia [22].

However, there is a dearth of information on the specific predictors of neonatal hypothermia in the first six hours after delivery in a local setting, as previous studies focused on the general predictors without considering the time frame. In addition, there is limited data on the preventive practices adopted for neonatal hypothermia by postnatal mothers, as prior studies focused on medical treatment and use guidelines rather than exploring the real practices implemented by postnatal mothers. Therefore, this study will address this gap by determining the predictors of neonatal hypothermia within six hours of birth and exploring preventive practices among postnatal mothers in the Kilimanjaro region.

## Broad objective

To determine the predictors of neonatal hypothermia within six hours of birth and exploring preventive practices among postnatal mothers in the Kilimanjaro region.

## The objective of the first phase: Quantitative study

1. To determine the prevalence of neonatal hypothermia within six hours of birth among neonates in the Kilimanjaro region.

2. To assess health facilities for the availability of equipment for preventing neonatal hypothermia in the Kilimanjaro region

3. To determine the factors associated with neonatal hypothermia within six hours of birth among neonates in the Kilimanjaro region

## Objective of the second phase: Qualitative study

1. To explore neonatal hypothermia prevention practices among postnatal mothers in the Kilimanjaro region

### Objective of the third phase

1. To develop strategies to increase adherence to thermal care guidelines to reduce neonatal hypothermia among post-natal mothers.

## Materials and methods

A mixed method with an explanatory sequential design and a deductive approach for data collection and analysis will be used. The aim of using a mixed method paradigm is based on the principles and logic of pragmatism. According to this paradigm, the mixed-use of quantitative and qualitative approaches results in a better understanding of the problem [23–25]. The quantitative data will be collected and analysed first in phase one, followed by qualitative data in the second phase, in which the quantitative findings will inform the second phase of the study. The quantitative and qualitative findings will be mixed in the data interpretation stage. The visual study diagram illustrating the research procedure is presented in Fig 1 below.

### Phase one: Quantitative study

First, a hospital-based analytical cross-sectional design will be used to determine the predictors of neonatal hypothermia within six hours of birth. The researcher opted for this design because

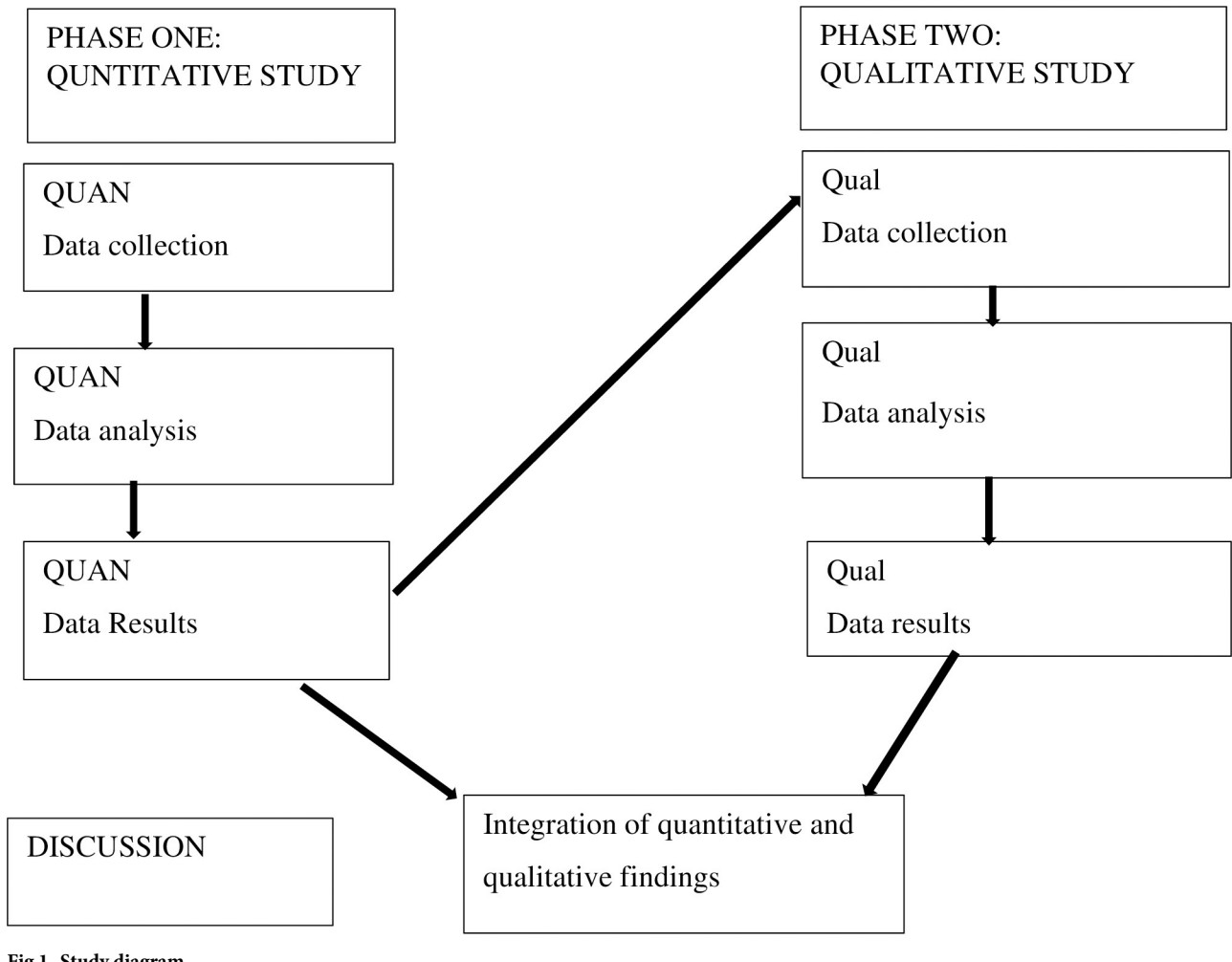

**Fig 1. Study diagram.**

it allows for the quantification of any potential association between exposures and neonatal hypothermia as well as the determination of the prevalence of neonatal hypothermia.

## Sample size and sampling methods

The sample size was calculated to be 293 neonates and their mothers by using Kish and Leslie (1965), who used a single population proportion formula based on a previous study conducted in Dar Es Salaam, Tanzania, which revealed a prevalence of 25.6% [12], a margin of error (d) of 0.05, an α = 0.05, and a 95% confidence interval and power of 80%. By considering a nonresponse rate of 10%, the final sample size was determined to be 325 neonates and their mothers. This study will be conducted in four health facilities in the Kilimanjaro region

$$n = \frac{(Z \propto /2)^2 \, P(1-P)}{d^2}$$

Whereby
n = sample size
$Z\alpha/2$ = 1.96 for 95% CI
d = Margin of error between the sample and population
P = Prevalence of neonatal hypothermia in neonates (25.6%)
from a previous study conducted in Tanzania [12]
Then

$$n = \frac{(1.96)^2 \text{ x } 0.256 \text{ x } (1-0.256)}{0.05^2}$$

n = 293 neonates

By considering 10% of non-response rate (r), then the final minimum sample size will be 325 neonates and their mothers

**Sampling technique.** The sampling technique will include multimethod sampling whereby different sampling techniques will be used in various stages. The first stage will involve the selection of two districts out of seven districts of the Kilimanjaro region using simple random sampling with lottery methods. The second stage involved the selection of two primary health facilities per district out of 144 primary health facilities. Moreover, the two districts were selected out of seven districts of the Kilimanjaro region because all districts share similar characteristics in terms of social structure and culture. Additionally, four health facilities were selected because of the equal availability of healthcare and a similar number of facility delivery thus sample size can be achieved at a minimum of four facilities. The third stage will involve proportionate stratified sampling to allocate several participants from each selected health facility, as proportional stratified sampling is used when a stratum sample size is allocated in proportion to the population size within the stratum. Last, the fourth stage will be systematic sampling, which will be used to select the number of neonates per day until a daily sample is obtained. The $K^{th}$ interval will be calculated using the formula $K^{th}$ = N/n [26], where N is the sampling frame, which is the total number of neonates admitted and n is the number of required samples per day.

## The eligibility criteria

### Inclusion criteria.

- Term and preterm neonates within 6 hours of birth (Preterm and full-term neonates will be included because we need to get diversification of the magnitude of the problem between

preterm and full-term neonates which may allow the development of evidence-based intervention and recommendations tailored to the unique needs of each population

• Postnatal mothers who agree to participate

**Exclusion criteria.**

• Neonates whose mothers will not be present

• Neonates whose mothers are in critical condition

## Scales and data collection

Quantitative data will be gathered by using four distinct methods:

**An interviewer-administered questionnaire.** This method will be used to obtain information from postnatal mothers. The information that will be obtained by this method is the biographical characteristics of postnatal mothers and behavioural characteristics associated with neonatal hypothermia.

*Procedure*. Four research assistants who work in neonatal intensive care units, operating theatres, labour wards, or postnatal wards will be trained for two days before data collection to ensure that they are familiar with the research procedure and will be monitored and guided by the principal researcher to maintain data quality control. Before data collection, the purpose of the study, benefits, and potential risks of the study will be explained to the subjects, and those who meet the inclusion criteria and are willing to participate will sign a consent form. The procedure will be performed in a private room to maintain confidentiality.

**Documentary review.** This method will be used to obtain information on neonatal and obstetric characteristics associated with neonatal hypothermia, such as the Apgar score, neonatal resuscitation history, neonatal medical problems or complications, mode of delivery, gestational age at birth, birth weight, time of delivery, and maternal obstetric complications.

*Procedure*. This method will also involve four research assistants, and the data that will be extracted will be from postnatal mothers' case files, which will be delivered within six hours of the data collection period.

**Observation method.** This method will be used to gather data on the availability of equipment in health facilities for preventing neonatal hypothermia, such as radiant warmers, incubators, thermostats, wall thermometers, plastic bags, polythene bags, and heated gel mattresses.

*Procedure*. The principal investigator and research assistant will assess the availability of equipment for preventing neonatal hypothermia through physical observation using a checklist.

**Measurement method.** A standardised digital thermometer will be used to measure the axillary temperature of neonates.

*Procedure*. The axillary body temperatures of the neonates will be measured by four research assistants, who will receive supportive supervision from the principal investigator. The equipment will be pretested for functionality and will be compared against the thermometers used by the health facilities one week before the actual data collection. Moreover, before the procedure, room temperature will be controlled by ensuring that the doors and windows are closed, and postnatal mothers will be informed about the research's objectives, advantages, and risks before data collection. Following approval and agreement to allow their neonates and themselves to be a part of the research, mothers will be instructed to hold their neonates in the appropriate position for axillary temperature, whereby the neonate axilla will be assessed. Then, a digital thermometer will be disinfected with 70% ethyl alcohol disinfectant, and the

thermometer will be placed in the high axilla, holding it against the side of the neonate until the beep sound is heard. The temperature will be recorded in the case file and by the data collection tool. Finally, the digital thermometer was disinfected with 70% ethyl alcohol during each procedure to prevent cross-contamination. The axillary temperature will be recorded from the first, third and sixth hour after delivery.

### Data collection tool

This study employs four data collection tools:

**Paper-based structured questionnaire.** A paper-based structured questionnaire was used to obtain data regarding behavioural characteristics associated with neonatal hypothermia. The tool will be adapted from (Demissie et al., 2018; Gendisha et al., 2019) and will be modified by the researcher to fit the study's local context. The tool has 22 items and will be organized into two sections: Section A will cover the socio-demographic characteristics of the postnatal mothers, which will have 7 items, and Section B will be composed of 13 items regarding behavioural characteristics.

**Documentary review guide.** This is also another tool that can be used to obtain information associated with neonatal hypothermia. The information that will be obtained by this method will include neonatal characteristics such as the Apgar score, birth weight, gestational age at delivery, sex, and age of neonates after delivery in hours, obstetric characteristics such as mode of delivery, pregnancy type, obstetric complications, ANC contacts, and environmental factors such as time of delivery and temperature of the environment.

**Observation checklist.** This tool will be used to collect information on the availability of equipment for preventing neonatal hypothermia in health facilities. The tool consists of 10 items.

*Standardized calibrated digital thermometer.* A standardized calibrated FT 09 digital thermometer (Beurer-brand name), manufactured by Beurer GmbH, Germany, can measure 32.0˚C to 42.90˚C (89.60F to 109.90F) and has an accuracy of ±0.1˚C for the temperature range of 35.5–42.0˚C and ± 0.2˚C for the temperature range of 32.0–35.5˚C or above 42.0˚C. This standardized digital thermometer will be sourced from a supplier accredited by the health facility procurement department, will be pretested one week before actual data collection for functionality and will be compared to the digital thermometer that is used by health facilities. Moreover, the serial axillary temperature of the neonates will be taken at the 1st, 3rd and 6th hours.

### Validity and reliability

**Validity.** Both the face and content validity will be ensured. For face validity, the tool will be piloted using 10% (n = 33) of the required sample of different study subjects with the same characteristics but of different geographical locations to check if the tool is clear, understandable, answerable, language clarity, duration, and cultural acceptability. Also, the tool will be translated from English into Swahili to blend in with the study subjects understanding level. Furthermore, for content validity, the tool will be subjected to supervisors, statisticians and experts on the topic under study to check whether it adequately covers all domains related to the variables and if it answers research questions. These individuals will provide feedback on items such as organization, language, structure of the sentence and adequate of the items to answer research questions. Thereafter the tool will be returned to them for their final verification until there are no more inputs that need to be added.

### Reliability

To ensure the reliability of the tool, a test-retest will be employed in which the tool will be tested and retested on different occasions with similar populations to see the consistency of the

results and also for internal consistency of the tool, Cronbach's alpha coefficient will be calculated after data collection using 10% of the sample required. In this study, a Cronbach's alpha coefficient of 0.70 and above will be considered acceptable, and scale analysis will be carried out specifically principal component analysis (PCA) will be done if the Cronbach's alpha coefficient does not reach the criterion level of alpha = 0.7.

**Variable measurement.** *Dependent variable*. Neonatal hypothermia will be measured by one item on an ordinal scale to determine whether the axillary temperature is measured within six hours at the 1st, 3rd and 6th hours after delivery for 3 minutes or until the beep sound is heard, which is less than 36.50˚C, by taking the axillary temperature of neonates using a standardised digital thermometer. A temperature reading of 36.5˚C to 37.5˚C will be considered normal, a temperature reading of 36.0˚C–36.4˚C will be considered mild hypothermia and will be coded as 1, a reading of 32.0˚C–35.9˚C will be considered moderate hypothermia and was coded as 2, and an axillary temperature reading < 32.0˚C will be considered severe hypothermia and will be coded as 3.

*Independent variables. Maternal social demographic characteristics*: This variable will be measured by 7 items, in which maternal age will be measured by an interval scale; marital status, tribe, residence, occupation and religion will be measured by a nominal scale; and education level will be measured by an ordinal scale.

*Neonatal characteristics*: This will be measured by 7 items: birth weight, age of neonates in hours, which will be measured by an interval scale; gestational age at delivery, Apgar score; and resuscitation history, which will be measured by a binary scale, while the sex of neonates will be measured by a nominal scale and axillary temperature will be measured by an ordinal scale.

*Obstetric characteristics*: This variable will be measured by five items on binary and interval scales: mode of delivery, parity, pregnancy type, and obstetric complications (binary scale), while the number of ANC visits will be measured on an interval scale.

*Behavioural characteristics*: This will be measured by 9 items with a binary scale on whether postnatal mothers perform skin-to-skin contact practices, immediately dry neonates, cover their head with a cap, wear socks, initiate breastfeeding within one hour, weigh postponement for at least 60–90 minutes, bathed a neonate after 6 hours, rooming practices, resuscitation, warm transportation of a neonate, and place of delivery.

*Environmental characteristics*: This variable will be measured by 2 items on a binary scale on time of delivery and temperature of the environment. The time of delivery will be either daytime or nighttime (those neonates who will be delivered from 6 A.m. to 6:00 P.m. will be considered to be delivered during the daytime, and those neonates who will be delivered from 6:01 P.m. to 5: 59 A.m. will be considered to be delivered at night time). The temperature of the environment on the day of delivery will be hot, warm or cold.

**Data analysis.** Quantitative data will be analysed with the SPSS-20 package. This will occur after data cleaning through the running frequency to ensure that all the data are captured. The sociodemographic characteristics of postnatal mothers and obstetric characteristics will be determined by frequency and percentage through descriptive statistics. The prevalence of neonatal hypothermia will be determined by frequency and percentage through descriptive statistics in which the frequency distribution of axillary temperature readings, proportion of neonatal hypothermia, and categories of neonatal hypothermia (mild, moderate and severe) will be reported together with comparative observations of neonatal hypothermia in three observations. Moreover, the availability of equipment for preventing neonatal hypothermia will be determined by frequency and percentage. Finally, the relationships between postnatal maternal characteristics, neonatal characteristics, obstetric characteristics, behavioural characteristics, and environmental characteristics were determined by the chi-square test through

inferential statistics. The associations between postnatal maternal characteristics, neonatal characteristics, obstetric characteristics, behavioural characteristics, and environmental characteristics will be determined by binary logistic regression.

## Phase two: Qualitative study

The second phase consisted of an explorative qualitative study that used a descriptive qualitative design to explore preventive practices for neonatal hypothermia among postnatal mothers. The study will opt for this design because it provides a direct description of participants' experiences without a deep theoretical context of a particular topic under investigation, and the design is flexible and has sufficient procedures to provide a thorough understanding of the phenomenon [27]. This qualitative approach builds upon findings collected from the quantitative phase whereby postnatal mothers whose neonates will be identified as hypothermic or normothermic in the quantitative phase will be interviewed in this phase based on saturation points and aims to provide a deeper understanding of neonatal hypothermia. Thematic analysis will be employed to analyse the qualitative data that will be collected.

## Researcher characteristics and reflexivity

The principal investigator is a midwifery student with 8 years of experience in clinical practices and training institutions and has managed to work in the labour ward, postnatal ward, reproductive unit, paediatric ward, neonatal ward and training institution. Through this experience, the researcher is expecting to perform an extensive exploration of preventive practices for neonatal hypothermia among postnatal mothers. Moreover, the principles of conducting interviews in a qualitative study, such as bracketing, use of reflexive journals and taking field notes, will be ensured to prevent researchers from influencing the interviews.

## Sampling method

Criterion purposive sampling will be used to recruit and select participants. Participants will be selected based on their ability to express themselves and if their neonates are hypothermic or normothermic in the quantitative phase. The principal investigator will ensure maximum diversity in background characteristics such as age, level of education, culture, religion, and economic status during selection and recruitment to gain a comprehensive understanding of preventive practices. In the qualitative study, there is no fixed sample size, the estimation of sample size is guided by the principle of saturation point or information power, and in this study, and data saturation point will be reached when there are no more new points with emerging concepts or insights from further interviews. However, for planning purposes, 15 postnatal mothers will be interviewed and the replacement of the participants will be used to maintain the sample size in case of withdrawal. In addition, before the discussion, all participants will be given written informed consent for their participation.

## Data collection method

**In-depth interview.**   A face-to-face in-depth interview will be used to collect qualitative data on preventive practices for neonatal hypothermia among postnatal mothers. The researcher has opted for this method because the study intends to explore the individualised and not shared preventive practices of neonatal hypothermia among post-natal mothers.

*Procedure*. The in-depth interview will be conducted in a private, conducive and friendly room where the participants will be comfortable to express themselves. The interviews will commence after the researcher to explain the purpose, benefits, risks, voluntary participation,

privacy and confidentiality, informed consent will be obtained from the participants and the participants will be given codes to protect their identities. Each interview will start with a general question to understand the general experiences of participants and will be followed by several probing questions to gain a breadth and deeper understanding of the preventive practices of neonatal hypothermia. The interviews will be recorded using an audio-recorder and field notes will be used to record verbal cues, the interview duration will depend on the participant's tolerance and patience to describe their experiences under the topic interviewed. However, the interview will range from 45–90 minutes and the key issues will be summarized immediately after every interview to be conducted.

## Data collection tool

A qualitative interview guide will be used to collect data on preventive practices for neonatal hypothermia among postnatal mothers. The tool was designed by the principal investigator based on the research question. The semi-structured interview guide consisted of 3 main interview topics, which were constructed to explore into details of this phenomenon, allowing the participants to express their experiences that might be overlooked when closed-ended questions were used. Additionally, an audio recorder will be used to capture and preserve audio from the interview. It is chosen because it ensures the accuracy of the data in the participants' responses, accuracy in recording sounds, long-lasting battery life, large storage, easy operating, simple to carry and user-friendly to fieldwork. In addition, the recorder includes backup functions to protect against data loss in the event of unanticipated occurrences or technical issues. The confidentiality and integrity of the data will be guaranteed by the implementation of security measures, such as encoding and password protection. Furthermore, the audio recorder will be pre-tested before actual data collection to check for functionality, ability to transfer data, and capturing of sounds, in terms of, audibility which all of these will ensure the credibility of the tool to be used in this study.

## Data analysis

Thematic analysis will be used to analyse the data on preventive practices for neonatal hypothermia among postnatal mothers. Thematic analysis is chosen because it provides a purely qualitative account of richer and more detailed data. Data will be collected using audio recorders and field notes whereby audio recordings will be transcribed verbatim and translated to English using a forward and backward translation approach. The thematic analysis will follow the following steps according to Braun and Clarke (2006) which are (i) familiarization; This is the first stage at the manifest level of thematic analysis that involves transcribing the data from audio to text, reading and re-reading the data, identifying the initial ideas about the interviewed topics to familiarize with the collected data. At this step the researcher immerses in the data to identify the key concept within the data set. (ii) **Generating initial codes**: This will be the second stage after familiarization which will involve coding the interesting and peculiar characteristics of the data systematically across the entire data set, assembling data relevant to each code. The researcher will identify the meaningful units of text, in the form of phrases which will be highlighted with different colours (iii) **Searching for themes**: The stage will involve organizing codes from different discussants into potential themes by mixing many codes to generate a single theme. The researcher will identify patterns among the concepts and begin to develop themes according to the coded patterns. (iv) **Reviewing themes**: The researcher will review and refine the themes to ensure if are accurate and relevant to the data. Themes will be compared to the data to ensure truly presentation of the data from the Discussant. (v) **Defining and naming of themes:** After identifying the final themes, the researcher

will define and name the themes in a way that it presents the real meaning of the phenomenon. The themes' names will be examined to make sure they are sufficiently descriptive and brief to be included in the report. (vi) **Producing report and writing up**: This will be the final stage of thematic analysis that will involve writing the report using the analysed data. The report will comprise a narrative description and quotations from participants.

## Trustworthiness

Trustworthiness and authenticity are taken into consideration through the following elements:

Credibility will be ensured through member checks, persistent observation, and prolonged engagement. Additionally, the interviews will be audio recorded and then transcribed verbatim to ensure that each detailed piece of information provided by the participants is captured and analysed. Dependability will be guaranteed by purposively selecting participants, employing triangulation and using an interview guide, two methods of data collection and field notes. Confirmability will be ensured by participants' validation of transcripts (member checking) and the use of participants' verbatim quotes to present the results and by practising researcher bracketing. Moreover, participants' nonverbal cues will be captured in field notes. Transferability will be ensured through writing a study setting, participant characteristics that will represent a general population, and themes and subthemes that will allow the applicability of these findings to another similar setting.

## Ethical consideration

Ethical approval was obtained from the institutional research review ethics committee (IRREC) of the University of Dodoma with reference number MA.84/261/69/2. Permission to conduct this study at the selected health facilities will be obtained from the permanent secretary president's Office, the Regional Administration and Local Government (PO-RALG) and the Regional Administration Secretary (RAS) of the Kilimanjaro region. Informed consent will be obtained after the participants are informed about the purpose of the study, benefits, risks, voluntary participation and ability to withdraw at any time during the study. Privacy and confidentiality will be ensured by assigning numbers to participants for anonymity, and information that will be obtained will be kept safe. Moreover, hypothermic neonates, who will be identified during the study period will be reported to the ward in-charge of the respective unit for appropriate intervention and will be intervened based on their severity, and their mothers will be advised to adhere to thermal care while at health facilities and home.

## Discussion

Neonatal hypothermia is a public concern that contributes to neonatal health impairment and neonatal mortality. Globally, 2.3 million children die in the first 28 days after delivery, and approximately 6,400 neonates die every day; one of its contributors is neonatal hypothermia [2, 4]. Despite the use of different strategies to reduce the neonatal mortality rate, neonatal mortality in sub-Saharan African countries was estimated to be 27 deaths per 1000 live births in 2019, which is still high, and in Tanzania, the neonatal mortality rate is expected to reach 24 deaths per 1000 live births in 2022 [20].

The neonatal period from 0 to 28 days is a critical period for neonatal survival in which hypothermia is one of the key contributing factors to poor growth and development, poor feeding, hypoglycemia, restlessness, respiratory distress and infection, all of which contribute to adverse outcomes such as neonatal mortality. Previous studies have reported this problem, but there is a paucity of evidence on the factors associated with neonatal hypothermia at specific times, the availability of thermal equipment in health facilities and preventive practices

for neonatal hypothermia among postnatal mothers. This mixed-method study will provide a wide understanding of this problem, which will lead to the recommendation of different interventions for preventing neonatal hypothermia.

## Duration of the study

The study is expected to be conducted from April 2024 to December 2024 in the Kilimanjaro region, Tanzania in the selected health facilities

## Strengths and limitations of the study

This proposed study is going to provide insight into neonatal hypothermia in the region by providing the magnitude, factors associated with it, thermal equipment available in the selected health facilities, and real practices of post-natal mothers on preventing hypothermia, which will provide baseline information for proposing effective, costless, and friendly interventions to prevent neonatal hypothermia. Moreover, by utilising a mixed-method sequential explanatory design, the study will be able to quantify and be supplemented with qualitative data to gain an in-depth understanding of the phenomenon, which most of the previous studies didn't do. Despite the stated strengths, the proposed study has some limitations. The first limitation may be that the study will not be able to follow the neonates who will be hypothermic after six hours to determine their outcomes. Last but not least, the study population will be limited to the Kilimanjaro region, which is not generalizable to the entire Tanzania mainland, with 26 regions raising concerns about its external validity.

## Supporting information

**S1 File. In-depth interview guide for post-natal mothers.**
(DOCX)

## Acknowledgments

We acknowledge the University of Dodoma for education support, which resulted in the drafting of this protocol and providing ethical clearance for this study.

## Author Contributions

**Conceptualization:** Emmanuel Daniel.

**Methodology:** Emmanuel Daniel, Saada A. Seif, Walter C. Millanzi.

**Supervision:** Saada A. Seif, Walter C. Millanzi.

**Validation:** Emmanuel Daniel, Saada A. Seif, Walter C. Millanzi.

**Writing – original draft:** Emmanuel Daniel, Saada A. Seif, Walter C. Millanzi.

**Writing – review & editing:** Emmanuel Daniel, Saada A. Seif, Walter C. Millanzi.

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
