## [Decision Letter · Decision Letter 0]

9 Jul 2024

PONE-D-24-12746Predictors of neonatal hypothermia within six hours of birth and exploring preventive practices among post-natal mothers in Kilimanjaro region: explanatory sequential mixed method protocolPLOS ONE

Dear Dr. Daniel,

Thank you for submitting your manuscript to PLOS ONE. After careful consideration, we feel that it has merit but does not fully meet PLOS ONE’s publication criteria as it currently stands. Therefore, we invite you to submit a revised version of the manuscript that addresses the points raised during the review process.

We look forward to receiving your revised manuscript.

Kind regards,

Kahsu Gebrekidan, Ph.D.

Academic Editor

PLOS ONE

Journal Requirements:

Reviewers' comments:

Reviewer's Responses to Questions

**Comments to the Author**

1. Does the manuscript provide a valid rationale for the proposed study, with clearly identified and justified research questions?

Reviewer #1: Yes

Reviewer #2: Yes

2. Is the protocol technically sound and planned in a manner that will lead to a meaningful outcome and allow testing the stated hypotheses?

Reviewer #1: Yes

Reviewer #2: Yes

3. Is the methodology feasible and described in sufficient detail to allow the work to be replicable?

Reviewer #1: No

Reviewer #2: Yes

4. Have the authors described where all data underlying the findings will be made available when the study is complete?

Reviewer #1: Yes

Reviewer #2: Yes

5. Is the manuscript presented in an intelligible fashion and written in standard English?

Reviewer #1: Yes

Reviewer #2: Yes

6. Review Comments to the Author

You may also provide optional suggestions and comments to authors that they might find helpful in planning their study.

Reviewer #1: More details about analysis of the qualitative data.

For generalizability and reproducibility, researchers must consider the study setting and district, as well as the sample size

finally revise the required and resubmit

Reviewer #2: this is a study with a strong design. there is no commenrt

this is a study with a strong design. there is no commenrt

this is a study with a strong design. there is no commenrt

this is a study with a strong design. there is no commenrt

this is a study with a strong design. there is no commenrt

7. PLOS authors have the option to publish the peer review history of their article (what does this mean?). If published, this will include your full peer review and any attached files.

Reviewer #1: **Yes: **Dr. Nahla Abdulrahman

Reviewer #2: No

---

## [Author Response · Author response to Decision Letter 0]

27 Aug 2024

Reviewer2: Author responses to comment 

Reviewer comment 1: The introduction is relevant and satisfactory information about the topic is presented in an organized way, yet, the researcher needs to add more Introduction is relevant and satisfactory information about the topic is presented in an organized way, yet, researcher needs to add more previous studies mainly related to prediction and preventive measures taken by mothers worldwide, not only in the African region., not only in the African region.

Response: Thanks for the comment, previous studies mainly related to prediction and preventive measures taken by mothers worldwide, have been added to the introduction (Pg 5 of revised protocol) 

Reviewer comment 2: On page no 3: I suggest using "implications of study” instead of discussion.

Response: Thanks for the suggestion, No, change has been made since due to PLOS one journal guideline, hence the word discussion remained in the abstract as appeared before (Pg 3 of the revised study protocol)

Reviewer comment 3: Also, the researcher uses the term (interview to collect the qualitative data). It is better to use the term (Focus group discussing: FGD) as a preferable technique for the collection of this type of data.

Response: This study intends to explore the individualized experience about the phenomenon and not a shared experience, therefore in-depth interview was opted for and not FGDs(Pg. 18-19 of the revised study protocol)

Reviewer comment 4: Page 6: Researchers must add objective(s) related to (protocol) that researchers seek to achieve and develop.

Response: To develop strategies to increase adherence to thermal care guidelines to reduce neonatal hypothermia among post-natal mothers (Pg. 6 of the revised study protocol)

Reviewer comment 5: The methods and materials are generally appropriate. However, more elaboration in the analysis of the qualitative data is needed, and researchers should provide justification

Response: More details about the analysis of the qualitative study have been revised (Pg. 20-21 of the revised study protocol)

Reviewer comment 6: Why only two districts were chosen and how many districts are there in the country under study, as well as why only four health facilities?

Response: Two districts were selected out of seven districts of the Kilimanjaro region because all districts share similar characteristics in terms of social structure and culture. Additionally, four health facilities were selected because of the equal availability of healthcare and a similar number of facility delivery thus sample size can be achieved at a minimum of four facilities (Pg. 9 of the revised study protocol)

Reviewer comment 7: Page no 8: more expansion why researchers chose both (full-term & preterm neonates) as samples, it is recognized that heat loss risk factors differ between premature and full-term neonates

Response: The researcher has selected both full-term and preterm neonates because of getting diversification of the magnitude of the problem and the development of evidence-based interventions and recommendations tailored to the unique needs of each population (Pg. 10 of the revised study protocol)

Reviewer comment 8: Data collection tools: Researchers need to add how to test the validity and reliability of data collection tools

Response: The reliability and validity of the tools have been revised in the study protocol (Pg.13-14 of the revised study protocol)

Reviewer comment 9: Researchers will take (15) post-natal mothers as samples, so they need to add what precautions will be taken in case of postnatal mothers’ withdrawal from the study. Some references need to be updated.

Response: The study will take 15 postnatal women as a baseline sample size since, in qualitative research, there is no fixed sample size; the sample size is determined by data saturation as indicated in the revised study protocol but for the case of withdrawal from the study, the replacement of the participants will be used as a mechanism to maintain the sample size in case of withdraw (Pg. 18 of the revised study protocol)

Reviewer comment 10: Some references need to be updated

Response: All References revised and all are updated (Pg 26-29 of the revised study protocol)

Editors’ comments 

Thanks for the constructive comments: The author(s) have addressed all comments from your office to abide by PLOS One journal guidelines and editor guidelines and all comments have been addressed in submission system and also in study protocol

---

## [Editor Report · Decision Letter 1]

24 Oct 2024

Predictors of neonatal hypothermia within six hours of birth and exploring preventive practices among post-natal mothers in Kilimanjaro region: explanatory sequential mixed method protocol

PONE-D-24-12746R1

Dear Mr. Emmanuel,

We’re pleased to inform you that your manuscript has been judged scientifically suitable for publication and will be formally accepted for publication once it meets all outstanding technical requirements.

Kind regards,

Kahsu Gebrekidan, Ph.D.

Academic Editor

PLOS ONE
---

## [Editor Report · Acceptance letter]

29 Oct 2024

PONE-D-24-12746R1 

PLOS ONE

Dear Dr. Daniel, 

I'm pleased to inform you that your manuscript has been deemed suitable for publication in PLOS ONE. Congratulations! Your manuscript is now being handed over to our production team.

Kind regards, 

on behalf of

Dr. Kahsu Gebrekidan 

Academic Editor

PLOS ONE